# Multidisciplinary In-Depth Investigation in a Young Athlete Suffering from Syncope Caused by Myocardial Bridge

**DOI:** 10.3390/diagnostics11112144

**Published:** 2021-11-19

**Authors:** Mariarita Brancaccio, Cristina Mennitti, Arturo Cesaro, Emanuele Monda, Valeria D’Argenio, Giorgio Casaburi, Cristina Mazzaccara, Annaluisa Ranieri, Fabio Fimiani, Ferdinando Barretta, Fabiana Uomo, Martina Caiazza, Michele Lioncino, Giovanni D’Alicandro, Giuseppe Limongelli, Paolo Calabrò, Daniela Terracciano, Barbara Lombardo, Giulia Frisso, Olga Scudiero

**Affiliations:** 1Department of Molecular Medicine and Medical Biotechnology, University of Naples Federico II, 80131 Naples, Italy; brancacciomariarita2@gmail.com (M.B.); cristinamennitti@libero.it (C.M.); cristina.mazzaccara@unina.it (C.M.); barretta@ceinge.unina.it (F.B.); fa.uomo@studenti.unina.it (F.U.); barbara.lombardo@unina.it (B.L.); 2Department of Translational Medical Sciences, Università degli Studi della Campania “Luigi Vanvitelli”, 80138 Naples, Italy; arturocesaro@hotmail.it (A.C.); emanuelemonda@me.com (E.M.); michelelioncino@icloud.com (M.L.); limongelligiuseppe@libero.it (G.L.); paolo.calabro@unicampania.it (P.C.); 3Division of Clinical Cardiology, A.O.R.N. “Sant’Anna e San Sebastiano”, 81100 Caserta, Italy; 4Ceinge Biotecnologie Avanzate S. C. a R. L., 80131 Naples, Italy; dargenio@ceinge.unina.it (V.D.); ranieria@ceinge.unina.it (A.R.); 5Department of Human Sciences and Quality of Life Promotion, San Raffaele Open University, Via di val Cannuta 247, 00166 Roma, Italy; 6Prescient Metabiomics, 1600 Faraday Ave, Carlsbad, CA 9200, USA; gcasaburi@prescientmetabiomics.com; 7Unit of Inherited and Rare Cardiovascular Diseases, Azienda Ospedaliera di Rilievo Nazionale AORN Dei Colli, “V. Monaldi”, 80122 Naples, Italy; fimianifabio@hotmail.it; 8Inherited and Rare Cardiovascular Diseases, Department of Translational Medical Sciences, University of Campania “Luigi Vanvitelli”, Monaldi Hospital, 81100 Naples, Italy; martina.caiazza@yahoo.it; 9Department of Neuroscience and Rehabilitation, Center of Sports Medicine and Disability, AORN, Santobono-Pausillipon, 80122 Naples, Italy; ninodalicandro@libero.it; 10Department of Translational Medical Sciences, University of Naples Federico II, 80131 Naples, Italy; daniela.terracciano@unina.it; 11Task Force on Microbiome Studies, University of Naples Federico II, 80100 Naples, Italy

**Keywords:** athlete, sport activity, laboratory medicine, heart bridge, genomic analysis, exome sequencing, oligogenic combination network

## Abstract

Laboratory medicine, along with genetic investigations in sports medicine, is taking on an increasingly important role in monitoring athletes’ health conditions. Acute or intense exercise can result in metabolic imbalances, muscle injuries or reveal cardiovascular disorders. This study aimed to monitor the health status of a basketball player with an integrated approach, including biochemical and genetic investigations and advanced imaging techniques, to shed light on the causes of recurrent syncope he experienced during exercise. Biochemical analyses showed that the athlete had abnormal iron, ferritin and bilirubin levels. Coronary Computed Tomographic Angiography highlighted the presence of an intramyocardial bridge, suggesting this may be the cause of the observed syncopes. The athlete was excluded from competitive activity. In order to understand if this cardiac malformation could be caused by an inherited genetic condition, both array-CGH and whole exome sequencing were performed. Array-CGH showed two intronic deletions involving *MACROD2* and *COMMD10* genes, which could be related to a congenital heart defect; whole exome sequencing highlighted the genotype compatible with Gilbert syndrome. However, no clear pathogenic mutations related to the patient’s cardiological phenotype were detected, even after applying machine learning methods. This case report highlights the importance and the need to provide exhaustive personalized diagnostic work up for the athletes in order to cover the cause of their malaise and for safeguarding their health. This multidisciplinary approach can be useful to create ad personam training and treatments, thus avoiding the appearance of diseases and injuries which, if underestimated, can become irreversible disorders and sometimes can result in the death of the athlete.

## 1. Introduction

Laboratory medicine and medical genetics have become disciplines necessary for monitoring athletes’ health [1,2,3]. It is known that intense physical activity can cause cardiovascular disorders [4,5], thrombotic events [6,7], muscle damage [4,8,9] and infections [10,11,12,13], mostly in predisposed subjects. Consequently, the observation of the health status of competitive athletes is necessary for evaluating the stress induced by intense physical activity. Particularly, laboratory monitoring of athletes helps physicians and athletic trainers in the development of targeted training and recovery programs to reduce injuries and/or permanent damage [1,2,3,14]. Recently, in order to investigate athletes’ state of health, the scientific community has identified biomarkers useful as an alarm bell for the protection of the athlete [15] in order to early highlight urinary infections [16], activation of the immune system [17], brain damage [18], muscle injuries [19,20,21], cardiac disorders [3,22], hormonal [23,24] and vitamin dysregulations [25,26].

In addition, another fundamental aspect for the correct clinical evaluation of athletes includes cardiovascular examination and diagnostic imaging capable of identifying cardiac pathologies that are potentially lethal [27,28] or muscle injuries [29,30], which would be difficult to evaluate at the medical/laboratory examination alone.

At the same time, it is known that some forms of hereditary/congenital heart disease or metabolic defects, which can cause serious accidents such as sudden death, may not be evident in the absence of family history and/or specific symptoms [31,32].

Hence, the need to identify all determinants that can define the phenotype of athletes is becoming clear, shedding light on the metabolic characteristics and the mechanisms of adaptation and response of the individual athlete to the stress caused by intense physical activity.

In this scenario, we present a case report in which a multidisciplinary approach is used to investigate the causes that influenced the health status of a competitive athlete suffering from repeated syncopes. Using biochemical and haematological tests and cardiovascular and genetic investigations, we highlight the injuries that resulted in the player’s withdrawal from competitive activity while preserving his life.

## 2. Materials and Methods

### 2.1. Ethical Approval

The study was conducted according to the ethical guidelines of Helsinki Declaration of the World Medical Association and was approved by the ethics committee (protocol 200/17) of the University of Naples Federico II. The athlete provided written consent to carry out biochemical laboratory tests and genetic analysis.

### 2.2. Cardiovascular Evaluation

The cardiovascular evaluation of the athlete included a two-step approach [33,34]:(1)Standard clinical evaluation, as a part of the pre-participation screening, which consisted of family and personal history, physical examination, electrocardiogram (ECG) and stress test, according to the current recommendations [34];(2)Additional clinical investigations in the presence of one or more clinical or instrumental markers suggestive of a pathologic condition potentially associated with an increased risk of sudden cardiac death during physical activity. The additional investigations included echocardiography, 24–48 ECG Holter, electrophysiological study, coronary computed tomographic angiogram (CTA) and exercise echocardiogram.

### 2.3. Clinical Laboratory Analysis

Blood samples of the athlete have been collected in two different phases of the agonistic season: the first blood samples were collected in September (0 month), in the preseason phase, and the second samples were collected in November (1 month after the start of the championship, during an episode of myocardial ischemia). A panel of clinical chemistry, hematology, coagulation and hormonal assays was analysed, which provided a general picture of the athlete’s health, providing information regarding metabolic and inflammatory status, hepatic and renal function, iron profile, muscle metabolism, endocrine system and hematological parameters. The blood samples were taken in the morning (8:00 a.m.) before training, after 72 h of rest. As a safeguard measure, blood, serum and plasma samples were frozen at −80 °C in case any analysis had to be repeated. We also collected the urine sample to evaluate pH, specific weight, colour, appearance, presence of proteins, glucose, ketones, bilirubin, hemoglobin, nitrite, leukocyte esterase, bacterial cells, squamous cells, leukocytes and erythrocytes. The urine samples were taken in the morning before training, according to standard procedure. Standard biochemical analyses were performed by using a standard serum analyzer in Architect c16000 (Abbott Diagnostics, Chicago, IL, USA). Red blood cells, leukocytes and platelet counts were performed through the Siemens Advia 2120i (Siemens Healthcare, Munich, Germany, EU) hematology analyzer. 

Coagulation analyses were performed according to the manufacturer’s recommendation using a standard plasma analyzer ACLTOP550^CTS^ (Laboratory Company, Inova Diagnostics, Inc. and Biokit, San Diego, CA, USA).

Serum cortisol concentrations were determined by immunoassay procedures through the Immulite 2000 analyzer (Cortisol Immunoassay kit; Siemens Healthiness, Erlangen, Germany); the measurement of vitamin D and thyroid hormones (fT3, fT4 and TSH) was performed by automated enzyme immunoassays in chemiluminescence using Liaison XL (Diasorin, Saluggia, Italy, EU) and ADvia Centaur (Siemens Heltineers, Erlangen, Germany, EU), respectively. All the procedures took place according to the manufacturer’s recommendation. 

The first morning urine sample was collected and analyzed within four hours of arrival using an automated urine chemistry analyzer (UC3500, Sysmex, Kobe, Japan) and a fluorescence flow cytometer (UF 1000i, Sysmex, Kobe, Japan) according to the manufacturer’s instructions. When necessary, we conducted an examination using an optical microscope.

All analysis was performed in triplicate in order to guarantee the accuracy of results.

### 2.4. Genetic Test: Array-Comparative Genomic Hybridization

Genomic DNA was extracted from peripheral venous blood by the Illustra Nucleon Genomic DNA Extraction kit (GE Healthcare, UK). High resolution array-Comparative Genomic Hybridization (a-CGH) analysis was performed according to the manufacturer’s protocols. DNA specimen was analyzed with the Human Genome CGH Microarray kit 4 × 180K (Agilent Technologies, Santa Clara, CA, USA), with an average space of 13 Kb, and allowing an average resolution of 25 Kb. The microarray was scanned on an Agilent G2600D scanner. Image files were quantified, and data were visualized by using Agilent’s Cytogenomics software (V.4.0.3.12).

### 2.5. Genetic Test: Whole Exome Sequencing and DNA Variants Analysis

A total amount of 1.0 μg of genomic DNA, quantified using the Qubit dsDNA BR Assay (Thermo Fisher Scientific, Waltham, MA, USA), was used as input for the following molecular analysis. Exome enrichment was performed by using the Agilent SureSelect Human All ExonV6 kit (Agilent Technologies, Santa Clara, CA, USA), following the manufacturer’s recommendations and using a specific index sequence to tag the sample univocally. First, genomic DNA was sheared by a hydrodynamic shearing system (Covaris, Woburn, MA, USA) to obtain 180–280 bp length fragments. Next, fragments ends were blunted through exonuclease/polymerase enzymes, and after a beads-based purification step, sequencing adapters were ligated to fragmented ends. The adapted fragments were specifically enriched using the exome probes. The captured library has been purified by using the AMPure XP beads (Beckman Coulter, Beverly, MA, USA), and quality was assessed by using the Agilent high sensitivity DNA assay on the Agilent Bioanalyzer 2100 system (Agilent Technologies, CA, USA). Finally, sequencing was carried out by using Illumina Sequencers, according to the manufacturer’s instructions (Novogene Service). 

After quality filtering and adapters removal, FASTQ files were mapped against the reference human genome sequence to carry out variants calling by using the Illumina Genome Studio tool. The obtained variant call format (VCF) file was further analyzed by using eVAI software v1.2 (enGenome) to highlight any pathogenetic variant and prioritized variants according to ACMG’s guidelines [35].

### 2.6. Genetic Test: Prediction of Disease-Causing Variant Combinations and Network Representation

The VCF file was leveraged to further explore potential disease-causing variant combinations using ORVAL v2019 [36], which works on the GRCh37/hg19 human genome assembly. The first step in ORVAL is a variant filtering procedure in order to select relevant variants. We selected the following filtering parameters: (i) a Minor allele frequency (MAF) threshold of 0.03 [37]; (ii) automatic removal of intergenic variants (i.e., variants that are not inside the defined gene coordinates based on hg19); and (iii) removal of intronic and non-splicing synonymous variants [36]. After the variant filtering step, a prediction step occurs in order to identify candidate disease-causing variant combinations. This step is performed with the Variant Combination Pathogenicity Predictor (VarCoPP) v1.0 tool [38]. VarCoPP uses Random Forest (RF) machine-learning methods to predict the pathogenicity of any bi-locus variant combination. Each predictor of VarCoPP has been trained on the pathogenic variant combinations present in the Digenic Diseases Database (DIDA) [39]. The predicted genes/variant are ranked by Gene Damage Index (GDI) [40]. Additionally, a machine-learning based method was also employed to predict digenic effect [41,42] of a pathogenic digenic variant combination identified and ranked by pathogenicity scores and confidence intervals [38]. For exploration purposes, the predicted oligogenic information was represented with gene networks in ORVAL to explore potential protein–protein interactions (PPI). The PPI network is built from the set of proteins belonging to the selected module by using the ComPPI database v2.1.1 [43]. Within the network, nodes represent genes, and edges connect two genes only if there exists at least one variant combination between them that has been predicted as candidate disease causing with VarCoPP. 

## 3. Results

### 3.1. Clinical History 

Herein, we report the case of a 19 year old competitive athlete who came to the emergency department for the occurrence of exercise-induced syncope without prodromes. He denied any family history of cardiovascular disease or sudden cardiac death in first-degree relatives. He was not a smoker and did not report the use of drugs or other medicaments. As advocated by current guidelines [34,44], he had been evaluated during routine pre-participation screening, which were unremarkable. During the examination, he reported a previous transient loss of consciousness at the age of 18, which was not investigated at that time.

At our department, he was conscious, and his vital parameters were within the reference range. Blood pressure and heart rate were measured while supine and during active standing for 3 min, as recommended at initial syncope evaluation [45], in order to exclude orthostatic hypotension. Continuous in-hospital ECG monitoring (Figure 1) did not show abnormalities, with the exclusion of asymptomatic sinus bradycardia. The echocardiographic evaluation showed mild left ventricular hypertrophy (LVH) with normal systolic and diastolic function. Exercise ECG showed enhanced QT dispersion in the absence of other abnormalities (Figure 1). Electrophysiologic study was performed; nevertheless, no inducible arrhythmia was detected. 

In consideration of the absence of cardiac abnormalities, participation into competitive sport was not discouraged.

After one year, the athlete experienced another syncopal episode during low-intensity exercise. Pulse assessment revealed tachycardia (180 bpm) that was unrelated to exertion; nevertheless, the patient was conscious and hemodynamically stable. The athlete was admitted to the emergency department, where baseline assessment did not show any abnormalities; however, routine blood samples revealed a slight increase in high sensitivity cardiac troponin (hs-Tn). Based on the concept that the probability of myocardial infarction increases with increasing hs-Tn values, the ESC 0/1 h algorithm to rule out acute coronary syndrome (ACS) was applied. Due to the fact that negative predictive value for patients meeting the “rule out” criteria approaches 99%, the diagnosis of non-ST-segment elevation myocardial infarction (NSTEMI) was excluded, and tachycardia was suspected to be the cause of myocardial injury. 

As a young athlete with mild LVH, cardiac magnetic resonance (CMR) was recommended to exclude possible underlying hypertrophic cardiomyopathy (HCM). At the same time, a blood sample was collected for the re-evaluation of biochemical-clinical and hematological data, as well as to conduct an in-depth genetic study with array-CGH and Whole Exome Sequencing (WES) analysis, which could highlight the presence of genetic variants with a higher risk of sudden cardiac death. However, CMR was not performed due to the patient’s refusal to undergo the exam. A coronary computed tomographic angiogram (CTA) was performed to exclude cardiovascular abnormalities and epicardial coronary stenoses. 

### 3.2. Instrumental Results 

Coronary CTA showed a long superficial bridge (20 mm) involving the middle segment of the left anterior descending (LAD) artery (Figure 2). In addition, the distal segments of diagonal arteries were involved. 

No atherosclerotic lesions were observed. Exercise echocardiogram was performed in order to document the presence of inducible ischemia, with evidence at the maximum heart rate of hypokinesia in antero-septal basal segments, which disappeared after exercise. The athlete received beta-blocker therapy and was withdrawn from competitions.

The opportunity to undergo surgical treatment for the myocardial bridge was considered; nevertheless, due to patient’s refusal, surgery was not performed. 

### 3.3. Clinical Laboratory Analysis 

The patient’s hematological and biochemical parameters were all in the reference range (see Appendix A), except those shown in Table 1. Particularly, martial homeostasis highlighted a condition of iron deficiency without anemia, which persisted even after two months of oral iron supplementary therapy, as evidenced by low serum ferritin in both samples (see Table 1). Total and indirect bilirubin increased at the first and second samplings, consistent with the known diagnosis of Gilbert’s syndrome. Hormones and vitamins levels evaluation underlined vitamin D hypovitaminosis and an aspecific slight elevation of FT3 at the first sampling.

### 3.4. Analysis of Array-CGH

By using array-CGH analysis, we detected a deletion of approximately 107.5 Kb on the 20 chromosome at p12.1 region (Figure 3), which includes partially intron 5 of *MACROD2* gene (RefSeq # NC_000020.11) and *MACROD2-AS1* gene. Furthermore, we identified a deletion of approximately 48.49 Kb on the 5 chromosome at q23.1 region, partially including the *COMMD10* gene (RefSeq # NC_000005.10) (Figure 3), and a deletion of approximately 38.4 Kb on the X chromosome at 13.3 region that partially includes the *MIR325HG* gene (RefSeq # NC_000023.11). 

The CNVs observed in this study were evaluated using the Database of Genomic Variants, the DECIPHER Database and the UCSC Genome Browser.

### 3.5. Exome Sequencing

Whole exome sequencing (WES) was performed in order to identify any pathogenic mutation potentially related to patient’s clinical phenotype. In total, 7.4 Gb was obtained, which is equivalent to more than 49 million reads, 99% of which passed quality filtering. Variants calling highlighted the presence of 271,966 variants that were further analyzed in order to verify the presence of clinically relevant variants. Variants were filtered based on coverage (>10X), variant’s allele frequency (>30%), a pathogenicity score based on ACMG criteria and ClinVar classification according to eVAI interpreter software pipeline.

Considering the pathogenicity score, six variants were identified (see Appendix A). All these variants are in a heterozygous state and have been associated to date to autosomal recessive disorders; moreover, no variants have been related to cardiovascular phenotypes.

In addition, another 103 variants were classified according to ClinVar in one of the following categories: pathogenic, likely pathogenic, drug response, association, risk factor, protective, affect and confers sensitivity (see Appendix A). However, none of these variants showed an association with the patient’s clinical signs. Thus, since exome analysis was not able to identify pathogenic mutations explaining the patient phenotype, we carried out a bioinformatic analysis to verify if the combination of more DNA variants may identify a risk haplotype. 

Furthermore, the analysis of the *UGT-1A1* gene promoter allowed genotyping the (TA)n polymorphism in the gene TATA box (rs3064744), highlighting that the athlete was heterozygous UGT1A1*1/*28.

### 3.6. Pathogenic Variant Predictions and Oligogenic Combination Network

In addition to ClinVar, a Random Forest (RF) approach was used to predict pathogenic variants using VarCopp. Sixty-five genes were classified with a median pathogenicity score ≥ 0.67 and represented in an interaction network (Figure 4A). A subnetwork for gene pairs carrying a severe pathogenicity score (i.e., ≥0.89) was also extrapolated (Figure 4B), resulting in 12 genes that were ranked based on their Gene Damage Index (GDI) (see Appendix A). Seven genes had unknown GDI. The gene with the lowest GDI and, thus, was more susceptible to disease-causing mutations was *FAM104B*, which translates to FAM104B (Family With Sequence Similarity 104, Member B). The *FAM104B* gene, located on chromosome X, has currently not clear known function but has been associated with both autistic disorder (PMID:25741868) and syndromic X-linked intellectual disability Lubs type (PMID:21681106; PMID:30208311). The second most susceptible gene to disease-causing mutation based on GDI score was *TBC1D8B* (TBC1 Domain Family Member 8B). Diseases associated with *TBC1D8B* include Nephrotic Syndrome Type 20 (PMID: 30661770) and Genetic Steroid-Resistant Nephrotic Syndrome (PMID: 31732614). Among its related pathways, Clathrin-derived vesicle budding and vesicle-mediated transport are observed. 

## 4. Discussion

The use of biochemical, haematological and genetic evaluation to assess risk factors in people who play sports is of growing interest at an amateur, competitive and elite level [1,3,6,46,47]. In this case report, we evaluated changes in biochemical, haematological and instrumental cardiovascular parameters and performed in-depth genetic analyses in order to shed light on the cause of repeated syncopes in an athlete who appeared to be in apparent good health but suffered from a few episodes of syncope. 

First of all, from the biochemical analysis, we highlighted a deficiency in the iron level. Competitive athletes, particularly teenage female athletes, may develop iron deficiency. Regular and above all high-intensity training increases iron losses by as much as 70% in athletes when compared to sedentary populations due to heavy sweating as well as increased blood loss in the urine and gastrointestinal tract [48]. However, insufficient dietary iron intake may also contribute to iron deficiency in athletes. One of the most sensitive and specific biomarker of iron deficiency is ferritin. Ferritin is the main iron storage protein within cells, and its concentration in blood reflects the extent of the mineral reserves in the body. Although ferritin levels less than 15 ng/mL are considered diagnostic for iron deficiency in the general population, a reasonable ferritin goal in athletes would be at least 30–40 ng/mL, as they need more iron than less active subjects. Our basketball player showed a condition of iron deficiency without anemia, which required oral iron supplementation and control blood tests after an at least 3-month long treatment. Measurement of Ab anti-transglut IgA (see Appendix A) ruled out that he was celiac. The check carried out after two months, following the second syncopal event, shows an improvement of serum iron and unsaturated iron binding capacity (UIBC) parameters, but there was still a significant reduction in iron deposits. The deficiency of iron contributes to cardiac and peripheral muscle dysfunction [49], emerging as a new comorbidity and a therapeutic target of chronic heart failure [49].

At the same time, biochemical investigations revealed that the athlete suffered from Gilbert’s syndrome, as evidenced by the condition of hyperbilirubinemia, with an increase in unconjugated bilirubin. Furthermore, sequencing by NGS allowed genotyping the (TA)n polymorphism (rs3064744) in the TATA box of the *UGT1A1* gene. TA insertions in the TATA box of the *UGT1A1* promoter are associated with hyperbilirubinaemia in Gilbert’s patients. Gilbert syndrome is a common hereditary condition characterized by mild hyperbilirubinemia. It is known that common genetic polymorphisms can be associated with the onset of diseases, including cardiovascular ones [50]. An insertional polymorphism of the TATA element upstream relative to *UGT1A1* gene results in reduced gene expression level [51]. Sequencing analysis showed that the athlete was heterozygous for UGT1A1*1/*28, which is compatible with the condition of moderate hyperbilirubinemia, and worsened following overexertion.

Moreover, the athlete showed vitamin D deficiency, particularly marked after a month of regular high-intensity training. Several authors indicated that vitamin D insufficiency is common among basketball players [52]. In addition, several reports are available demonstrating that Vitamin D deficiency among athletes was associated with increased Vitamin D receptor expression in skeletal muscles, resulting in high consumption of Vitamin D and low circulating levels [53,54]. Finally, the lower levels in the second sample (collected in November) compared to the first one (collected in September) may also be related to significant circannual rhythms of vitamin D [55]. Our findings support the idea that vitamin D circulating levels might represent a useful biomarker in athletes for assessing muscle activity levels.

In parallel, a slightly elevated FT3 but not of FT4 and TSH at the first sampling was also observed. This finding is in agreement with a previous report on hormonal changes after physique competition and did not reflect a pathological condition [56,57]. Normal serum TSH concentration, which is the most sensitive indicator of thyroid function, suggests an euthyroid status of the athlete [58,59]. We also observed decreased levels of FT3, FT4 and TSH after two months of physical activity. This finding was not surprising if we considered that exercise has an effect on body homeostasis (i.e., skeletal muscles and cardiac activity) involving the hypothalamus–hypophysis–thyroid axis [60]. Indeed, thyroid hormones and TSH levels changes are expected after physical exercise. Unfortunately, data on these variations were inconsistent, and it was difficult to achieve a definitive conclusion [61,62,63,64]. Thyroid hormone changes in athletes are still a matter of debate and need further investigations to assess their effect on athletes’ health status and performance.

Moreover, we found a decrease in serum testosterone in the trained athlete after 8 weeks of physical activity. This finding is in agreement with previous reports, showing that when athletes undergo an exercise program, testosterone significantly reduced regardless of body weight and sport [65,66,67]. 

In the meantime, instrumental investigations performed due to syncope episodes revealed the presence of a myocardial bridge. The myocardial bridge (MB) is a congenital abnormality characterized by an intramural course of a coronary artery. It was initially considered a benign condition, because the myocardium constricts the bridged coronary artery during systole, while blood flow occurs mainly during diastole. However, a possible association with myocardial infarction and sudden cardiac death was observed [68,69]. The risk of sudden cardiac death seems to be higher during strenuous physical activity. In particular, in the case of tachycardia, the blood supply of the myocardium becomes more dependent upon the systolic blood flow, which is impaired by the myocardial bridge. As a consequence, in the presence of a long myocardial bridge and prolonged physical activity, myocardial ischemia and exercise-induced life-threatening ventricular arrhythmias can occur. The most recent recommendations on sports cardiology suggest that athletes with MB and evidence of myocardial ischemia should be restricted from participation in competitive sports and be advised regarding leisure-time activities [70]. Cardiac CTA can provide a non-invasive anatomic assessment for MB given its high spatial resolution and unique ability to directly visualize coronary arteries, the myocardium, and their relationship. Moreover, it permits excluding concomitant atherosclerotic coronary artery disease and can demonstrate the relationship between the bridged artery and the adjacent branches, which is helpful in surgical planning. What is of importance is that the functional assessment of MB is required: the stress echocardiography recommended disqualification from competitive sports for ensuring the health and safety of the young athlete. Treatment with beta-blockers was prescribed as first-line therapy in the symptomatic athlete; although there was the indication to carry out surgery, it was not performed due to the patient’s refusal. 

Finally, we performed an in-depth genetic evaluation by using array-CGH and WES analysis [71,72] to verify the presence of a genetic background related to the patient’s cardiological phenotype. The array-CGH showed the presence of CNVs, which may be associated with the presence of congenital heart disorder. Notably, the macrodeletion in the *MACROD2* gene is worthy of attention. The protein encoded by *MACROD2* is a deacetylase involved in removing ADP-ribose from mono-ADP-ribosylated proteins. This gene is frequently involved in patients with complex syndromes. A recent genome-wide association study, including over 4000 patients affected by congenital heart disease and 8000 controls, revealed a statistically significant association between *MACROD2* polymorphisms and development of transposition of the great arteries, although the exact mechanisms remain unclear. Furthermore, data from Lahm et al. [73] show the expression of *MACROD2* in human embryonic cardiac cells; therefore, the authors speculated that this gene could act as a transcriptional regulator. 

Array-CGH analysis of the athlete showed a CNV alteration also affecting intron 5 of the *COMMD10* gene. *COMMD10* gene is involved in modulating the activity of the cullin-RING E3 ubiquitin ligase complexes (CRL) and reducing NF-kappa-B activation. In particular, it is known that some members of the family of 10 proteins containing the Murr1 domain of copper metabolism (COMMD) interacted with ENaC (epithelial channel of the sodium). COMMD10 likely contributes to the regulation of ENaC and may also be involved in long-term blood pressure control [74]. Another pioneering work proves that *COMMD10* was expressed in endothelial cells and smooth muscle cells of different tissues, assuming that COMMD10 may be closely related to angiogenesis in the embryo stage. In addition, the study revealed that *COMMD10* was expressed in the heart, manifesting that it is probably involved in heart development [75]. Thus, we could hypothesize a role for both *MACROD2* and *COMMD10* genes in the developing heart, thereby contributing, at least in part, to the development of congenital heart disorders, including MB.

On the other end, WES analysis was inconclusive, except for the definition of the (TA)n polymorphism in the *UGT1A1* gene. Therefore, we applied a machine learning method by using ORVAL v2019, which is a bioinformatic platform for the prediction and exploration of disease-causing oligogenic variant combinations. Recently, machine learning-based classification and prediction tools have been increasingly employed to predict rare genetic diseases, as well as to drive phenotype-informed genetic analysis. Artificial intelligence (AI) approaches have shown robust accuracy in estimating the risk of previously unappreciated genetic variants for common and complex diseases, including cancer and cardiovascular diseases [76,77]. Here, we tested whether machine learning techniques were able to predict underlying genetic conditions from NGS data. While the resulting models did not directly point to any known cardiac genetic conditions, a subnetwork of 12 gene pairs carrying a severe pathogenicity score (i.e., ≥0.89) was revealed. Among these, seven genes had unknown GDI, suggesting that the accumulated mutational damage of these genes in the healthy population could not be estimated with the existing data. This limitation is inherent to the field of AI, especially when applied to medicine, as such techniques heavily relies on training datasets such as the 1000 Genome Project database of gene variations [78]. This uncertainty of prediction power in newly introduced data is often caused by insufficient training data or an unbalance between the training and testing dataset distributions. Therefore, the majority of the proposed models usually collapse in the context of generalizability, failing to confirm predicting power in large-scale population studies [79]. This represents a fundamental limitation, making translational science from data to clinically use extremely challenging in the contemporary Big Data era. Nevertheless, recent efforts in combining multiple datasets, improvements in cloud-based computational power and the use of more sophisticated deep learning techniques may result in models that are clinically useful and could be potential tools for identifying hidden biomarkers and regulatory interactions in different disease conditions, making the use of AI a worthwhile effort to explore in genetic research [80].

## 5. Conclusions

In the modern conception of competitive sport, it is necessary to use a multidisciplinary approach to monitor athletes and to assess the stress that intense physical activity can cause.

Therefore, alongside the athlete’s medical history, it is now a duty for the sports medical community to combine biochemical and haematological tests that must be supported by instrumental evaluation in order to produce an overview of the athlete’s health status. In this scenario, genetic evaluation may have a relevant role when comprehensive clinical, laboratory and instrumental evaluations fail to identify a clear phenotype or to investigate the genetics and pathological mechanisms of a defined pathological condition. This in-depth analysis of the individual athlete can protect the sportsman during his/her competitive career, also contributing to a reduction in athlete’s sudden deaths.

In conclusion, our study aims to shed light on how sports medicine and genetics can support physicians and athletic trainers in understanding the athlete’s health, and at the same time it may be necessary for the identification of malformations/silent pathologies that could result in irreversible pathological conditions and/or the death of the athlete.

## Figures and Tables

**Figure 1 diagnostics-11-02144-f001:**
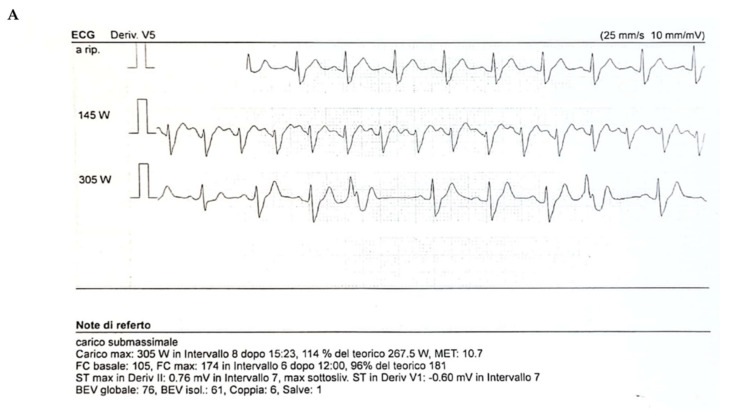
Representative images of the electrocardiogram. (**A**) Representation of the three electrocardiograms performed, the first at rest (Line 1) and the other two under exercise (Line 2 and 3). (**B**) Representative ECG cycles taken at rest (column 1) and during exercise (column 2 and 3).

**Figure 2 diagnostics-11-02144-f002:**
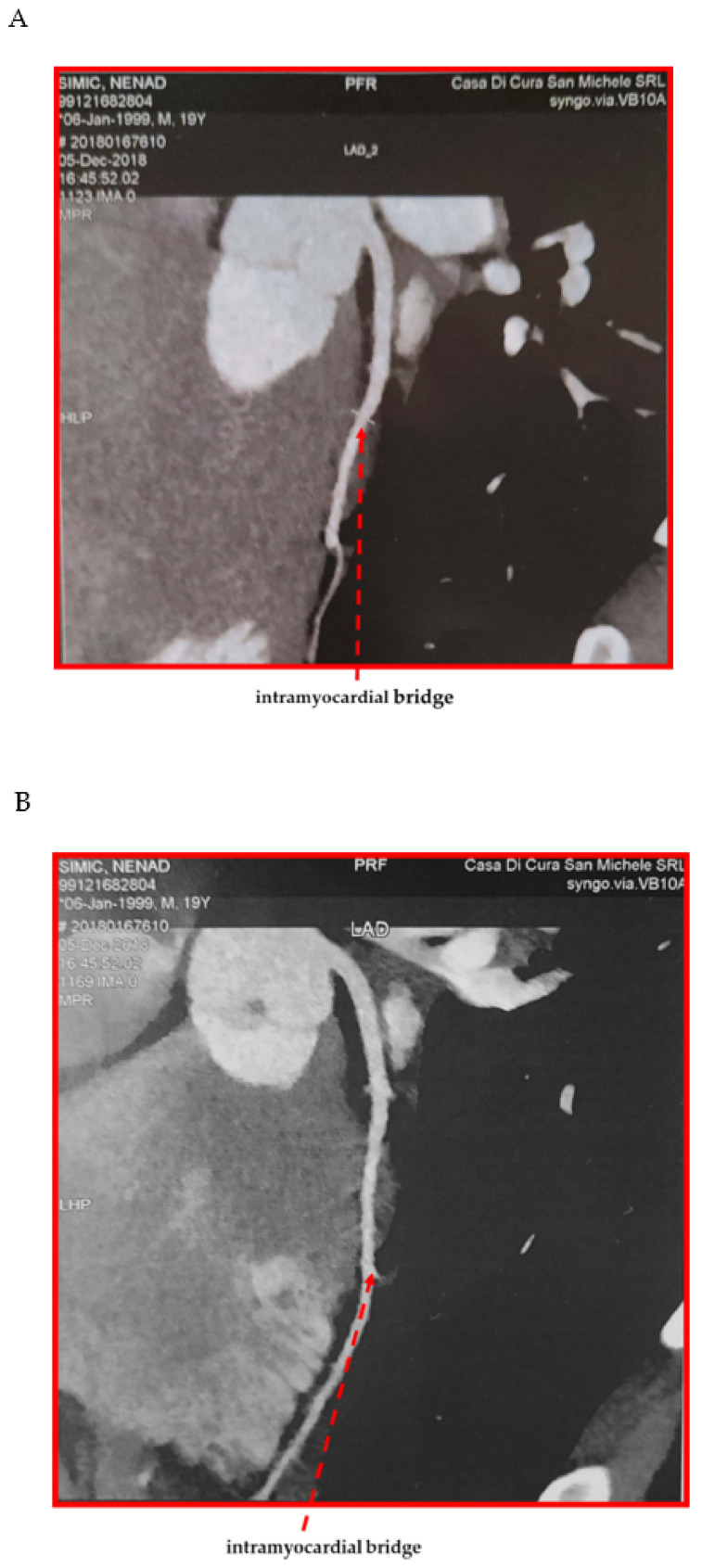
(**A**,**B**) Representative images of the intramyocardial bridge. (**C**) Examination performed with CTA and 128 × 2 by gating scanner, dual source, gantry rotation time 280 ms, collimation thickness 0.6mm, during injection of iodate i.v. with multiplanar reconstruction, cMPR (Multi Planar Reformation), MIP (Maximum Intensity Projection) and 3D volume rendering. Double reading exam. Reference standard for SCCT/SIRM international guidelines.

**Figure 3 diagnostics-11-02144-f003:**
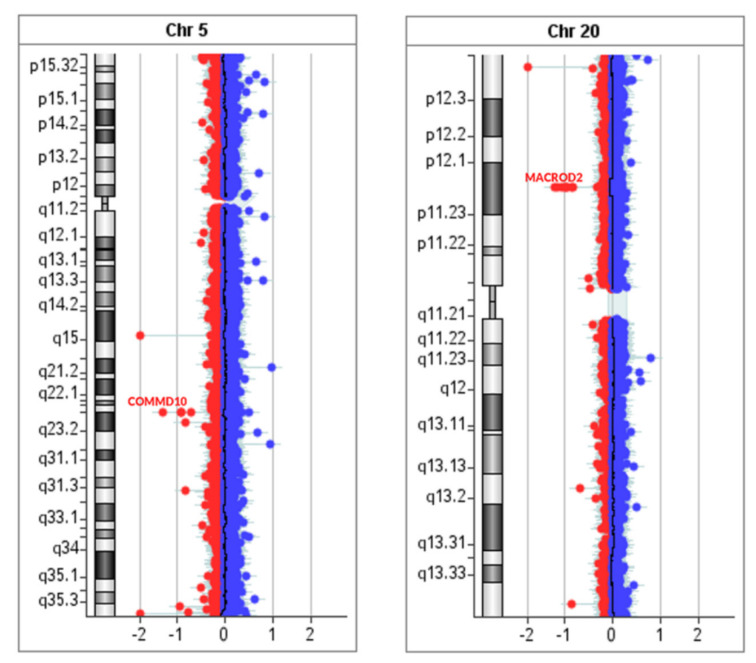
Array-CGH. The CGH analysis shows a heterozygous deletion on the 5 chromosome at q23.1 region, partially including *COMMD10* gene (**left**) and a heterozygous deletion on the 20 chromosome at p12.1 region, partially including *MACROD2* and *MACROD2-AS1* genes (**right**).

**Figure 4 diagnostics-11-02144-f004:**
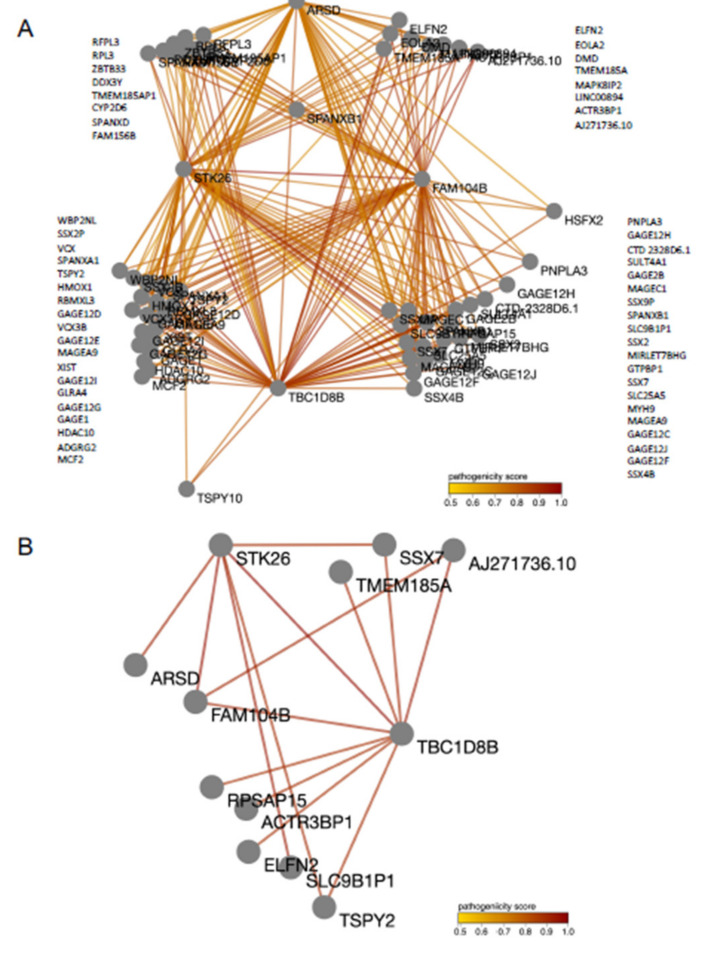
Oligogenic Combination Network. (**A**). Sixty-five genes had a median pathogenic score of ≥0.67. (**B**). Subset of network with highest median pathogenic score (≥89). Each node in the networks represent a gene reporting a pathogenic variant, while edges connect two genes only when there is at least one candidate disease-causing variant combination as predicted by VarCoPP. The colors of the edge represent the highest pathogenicity score for that pair and, more specifically, the highest Classification Score (CS) computed for a variant combination of that pair. This score is represented in a range from yellow (low pathogenicity score) to dark red (high pathogenicity score).

**Table 1 diagnostics-11-02144-t001:** Altered biochemical parameters found in the competitive athlete at time 0 (before the start of the championship) and time 1 (after 2 months from the start of the championship).

Reference Values	0 Month	2 Month
**Iron** (65–175 µg/dL) *	58	119
**UIBC** (69–240 µg/dL) *	309	220
**Ferritin** (22–275 ng/dL) *	10	10
**Total Bilirubin** (0.2–1.2 mg/dL) *	1.86	2.2
**Direct Bilirubin** (0–0.5 mg/dL) *	0.59	0.69
**Vitamin D** (>30 mg/dL) *	22.7	17.3
**FT3** (2.3–4.3 pg/mL) *	4.6	2.5

* Reference values of the Italian Society of Clinical Biochemistry and Clinical Molecular Biology (SIBioC).

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
