# Peer review of "Multidisciplinary In-Depth Investigation in a Young Athlete Suffering from Syncope Caused by Myocardial Bridge"

_diagnostics, 2021, doi:10.3390/diagnostics11112144_

Round 1

Reviewer 1 Report

Reviewer comments and suggestions

This study works out the health status of a basketball player through an integrated approach, including biochemical and genetic investigations and advanced imaging techniques, to confirm the recurrent syncope experienced during exercise. 

Biochemical analyses resulted in an abnormal iron, ferritin, and bilirubin levels. Coronary Computed Tomographic Angiography highlighted the presence of an intramyocardial bridge, proposing the cause of the observed syncopes. 

The study monitored the inherited genetic condition, using array-CGH and whole-exome sequencing. The result noted by the study was intronic deletions involving MACROD2 and COMMD10 genes, and other genotypes compatible with the Gilbert syndrome. The study recommends the importance of personalized diagnostic work-up for the athletes to cover the cause of their sickness, safeguarding their health. 

Decision: Major comments

Below are the comments for this paper to be incorporated in the revised version of the manuscript. 

  1. In the first introduction, the authors cited 42 papers that is not right. Better to reduce it. You also mention, hormonal [30-37] and vitamin dysregulations [38-42]. What was the need for citing many references here?
  2. Line 78-80 No need for these lines
  3. Table 1 legend should be in a full form which is written
  4. It is better to present your own result in the first para of the discussion
  5. Line 343 arrangement of references was not right. Needs to be changed (1-5).
  6. Line 361 exact function should be written here regarding ferritin
  7. Line 367-368 is there was any correction with the heart parameters
  8. One patient's report cannot be generalized, possibly a case of false positive or negative, How the author explain this situation (line 387-388)
  9. Line 466-467 How the authors perform
  10. Needed to point out the limitations of the study, I am a little bit worried that this paper could be a case report, not a full manuscript.
  11. Line 481-484 These lines were exaggerating as I could not find these types of data in the mentioned paper
  12. Check the format of reference number 12, 86,89,90

Author Response

Dear Editor,

thank you for the Reviewer’s Report about our manuscript entitled “Multidisciplinary in-depth investigation in a young athlete suffering from syncope caused by myocardial bridge”, submitted to Diagnostics. We have appreciated the comments received by the Reviewer and yourself and have carefully re-considered them in preparing a new version of the manuscript.

A point-by-point response to the comments is attached below.

We believe that the manuscript is now significantly improved thanks to the Reviewer’s inputs.

We hope that the new version of the paper deserve publication on Diagnostics.

Best regards,

Prof. Dr. Olga Scudiero and Prof. Dr. Giulia Frisso

Point-by-point response.

Reviewer 1

This study works out the health status of a basketball player through an integrated approach, including biochemical and genetic investigations and advanced imaging techniques, to confirm the recurrent syncope experienced during exercise. 

Biochemical analyses resulted in an abnormal iron, ferritin, and bilirubin levels. Coronary Computed Tomographic Angiography highlighted the presence of an intramyocardial bridge, proposing the cause of the observed syncopes. 

The study monitored the inherited genetic condition, using array-CGH and whole-exome sequencing. The result noted by the study was intronic deletions involving MACROD2 and COMMD10 genes, and other genotypes compatible with the Gilbert syndrome. The study recommends the importance of personalized diagnostic work-up for the athletes to cover the cause of their sickness, safeguarding their health. 

Decision: Major comments

Below are the comments for this paper to be incorporated in the revised version of the manuscript. 

Response:

First of all, we thank Reviewer 1 for comments and suggestions.

  • In the first introduction, the authors cited 42 papers that is not right. Better to reduce it. You also mention, hormonal [30-37] and vitamin dysregulations [38-42]. What was the need for citing many references here?

Thank you for your suggestion. Accordingly your consideration, in the introduction, we have reduced the references; in particular, we have eliminated 16 references. In particular, we have eliminated the oldest ones.

  • Line 78-80 No need for these lines

Accordingly your consideration, we have eliminated the lines.

  • Table 1 legend should be in a full form which is written

Accordingly your consideration, we rewrote the legend.

  • It is better to present your own result in the first part of the discussion

Accordingly your consideration, we have modified the first part of discussion in order to reduce the summery of knowledge in literature regarding this topic.

  • Line 343 arrangement of references was not right. Needs to be changed (1-5)

Accordingly your consideration, we modified the references

  • .Line 361 exact function should be written here regarding ferritin

Accordingly your consideration, we have specified the role of ferritin in the iron metabolism, lines 353-355.

  • Line 367-368 is there was any correction with the heart parameters

Accordingly your consideration, we have added the role of iron in heart damage lines 363-365 and, we have added a new reference number 49.

  • One patient's report cannot be generalized, possibly a case of false positive or negative, How the author explain this situation (line 387-388)

Accordingly your consideration, in the lines 139 paragraph 2.3 we have specified the procedure applied to perform the dosage of biochemical parameters in order to guarantee the accuracy.

In addition, in our previous work Mennitti el al,2020 reference number 4 and also, in literature was been demonstrated that in sport performed in door is normally to found a reduction of vitamin D level because is a micronutrient produced by sun exposition reference number 55. For this reason in the first dosage (September) we have found a normal level; after 2 month (November) we have found a reduction in vitamin D level.

  • Line 466-467 How the authors perform

The analysis has been performed through a bioinformatics approach using ORVAL v2019. It is reported in paragraph 2.4.3. Also, accordingly your consideration, we have specified in the discussion the type of approach used in our work in lines 457-458.

  • Needed to point out the limitations of the study, I am a little bit worried that this paper could be a case report, not a full manuscript.

Accordingly your consideration, we have changed in case report.

  • Line 481-484 These lines were exaggerating as I could not find these types of data in the mentioned paper

In our work we have described different methods to highlight the presence of genetic mutation e/o alteration; first we performed array-CGH, second we used NGS approach and finally thanks to Artificial Intelligence approach we found the genetic variant involved in cardiac disease. Following this, we wrote in line 481-484 now 476-480 the potential role of bioinformatics approach to support the identification of genetic variants.

We also, modified the text in order to clarify our sentences.

  • Check the format of reference number 12, 86,89,90

Accordingly your consideration, we have corrected the references format.

All modification was highlight in yellow in the text.

Reviewer 2 Report

The authors presented a manuscript as an original article on cardiovascular evaluation in athletes. The matter is worthy of investigation. In fact cardiovascular disease is the great pretender in apparently healthy subjects. Therefore, a deep analysis is mandatory in this setting.

Despite this promising introduction, the article miss the aim to reply questions the authors declared to study. In fact, after reading methods, I had a great expectation on results. On the contrary, I was extremely surprised to face results just on one patient. In this context there is a contrast between the first and the second part of the manuscript. Ultimately, this manuscript has to be considered as a care report more than an original article.

This is the main limitation of the paper.

Discussion is very interesting and appropriate to a case report, despite in some parts it appears different. In fact, indicating that they are studying basketball players suffering from syncope, again, the reader may expect a more complete manuscript, according also to the study protocol described in method. Unfortunately, reading the remaining discussion it is clear the nature of the manuscript.

Thus, considering all the previous comments, I suggest to reshape the manuscript as a case report.

I suggest also to include Electrocardiogram image

As minor issue, there are:

1-  in methods it is indicated many factories providing techniques to analyze the samples. However, it is indicate mostly the name of the factory. Some of them miss in the city or state where they are located. Please address.

2- in discussion, some references are indicated as the number of pubmed [PMID number]. I suggest to include in references as the other ones

3- a native english revision may help in increasing manuscript value

Author Response

Dear Editor,

thank you for the Reviewer’s Report about our manuscript entitled “Multidisciplinary in-depth investigation in a young athlete suffering from syncope caused by myocardial bridge”, submitted to Diagnostics. We have appreciated the comments received by the Reviewer and yourself and have carefully re-considered them in preparing a new version of the manuscript.

A point-by-point response to the comments is attached below.

We believe that the manuscript is now significantly improved thanks to the Reviewer’s inputs.

We hope that the new version of the paper deserve publication on Diagnostics.

Best regards,

Prof. Dr. Olga Scudiero and Prof. Dr. Giulia Frisso

Point-by-point response.

Reviewer 2

The authors presented a manuscript as an original article on cardiovascular evaluation in athletes. The matter is worthy of investigation. In fact cardiovascular disease is the great pretender in apparently healthy subjects. Therefore, a deep analysis is mandatory in this setting.

Despite this promising introduction, the article miss the aim to reply questions the authors declared to study. In fact, after reading methods, I had a great expectation on results. On the contrary, I was extremely surprised to face results just on one patient. In this context there is a contrast between the first and the second part of the manuscript. Ultimately, this manuscript has to be considered as a care report more than an original article.This is the main limitation of the paper.

Discussion is very interesting and appropriate to a case report, despite in some parts it appears different. In fact, indicating that they are studying basketball players suffering from syncope, again, the reader may expect a more complete manuscript, according also to the study protocol described in method. Unfortunately, reading the remaining discussion it is clear the nature of the manuscript.

Thus, considering all the previous comments, I suggest to reshape the manuscript as a case report.

Response:

First of all, we thank Reviewer 2 for comments and suggestions.

Accordingly your consideration, we have switched our manuscript in case report.

  • I suggest also to include Electrocardiogram image

Accordingly your consideration, we have added the Electrocardiogram image in the paragraph 3.1 figure 1a and 1b

  • As minor issue, there are:

1- in methods it is indicated many factories providing techniques to analyze the samples. However, it is indicate mostly the name of the factory. Some of them miss in the city or state where they are located. Please address.

Accordingly your consideration, we have added the information.

2- in discussion, some references are indicated as the number of pubmed [PMID number]. I suggest to include in references as the other ones

Accordingly your consideration, we have checked the references. Moreover, we didn’t add [PMID number] because is not required by journal guidelines.

3- a native english revision may help in increasing manuscript value

Accordingly your consideration, we have performed English editing by English editing office from Department of Molecular Medicine and Medical Biotechnology at University of Naples “Federico II”.

All modification was highlight in yellow in the text.

Round 2

Reviewer 1 Report

Please check line 202 

Confirm figure 1 A and 1 B

Reviewer 2 Report

The authors replied to all required issue. I suggest to accept in this version